# Mycobacterial biotin synthases require an auxiliary protein to convert dethiobiotin into biotin

Di Qu[1], Peng Ge[2], Laure Botella [1], Sae Woong Park [1], Ha-Na Lee [1], Natalie Thornton[1], James M. Bean [3], Inna V. Krieger[4], James C. Sacchettini [4], Sabine Ehrt [1], Courtney C. Aldrich [2] ✉ & Dirk Schnappinger [1] ✉

Lipid biosynthesis in the pathogen *Mycobacterium tuberculosis* depends on biotin for posttranslational modification of key enzymes. However, the mycobacterial biotin synthetic pathway is not fully understood. Here, we show that *rv1590*, a gene of previously unknown function, is required by *M. tuberculosis* to synthesize biotin. Chemical–generic interaction experiments mapped the function of *rv1590* to the conversion of dethiobiotin to biotin, which is catalyzed by biotin synthases (BioB). Biochemical studies confirmed that in contrast to BioB of *Escherichia coli*, BioB of *M. tuberculosis* requires Rv1590 (which we named "biotin synthase auxiliary protein" or BsaP), for activity. We found homologs of *bsaP* associated with *bioB* in many actinobacterial genomes, and confirmed that BioB of *Mycobacterium smegmatis* also requires BsaP. Structural comparisons of BsaP-associated biotin synthases with BsaP-independent biotin synthases suggest that the need for BsaP is determined by the [2Fe–2S] cluster that inserts sulfur into dethiobiotin. Our findings open new opportunities to seek BioB inhibitors to treat infections with *M. tuberculosis* and other pathogens.

With ~10 million people having fallen ill with tuberculosis (TB) in 2020, TB remains a major cause of sickness and mortality worldwide[1]. Even though significant progress has been made in developing new medicines for the treatment of TB[2–4], the incidence of this disease has been predicted to increase due to the disruption of health services by COVID-19[5]. The bacterium causing TB, *Mycobacterium tuberculosis*, is surrounded by a complex, lipid-rich cell envelope, which provides protection against killing by antibiotics and the immune system[6]. The various lipids that lend the mycobacterial cell envelope its selective permeability are synthesized from malonyl coenzyme A (CoA) building blocks, which are generated by several acyl-CoA carboxylases[7], each of which needs to be activated via posttranslational biotinylation, catalyzed by biotin protein ligase (BPL). Consequently, lipid biosynthesis depends on biotin, also known as vitamin $B_7$ or vitamin H. Partial

chemical inhibition of BPL increased the potency of two first-line drugs, rifampicin and ethambutol, and partial genetic interference with protein biotinylation accelerated clearance of *M. tuberculosis* from mouse lungs and spleens by rifampicin[8]. Furthermore, genetic inactivation of BPL can be sufficient to sterilize *M. tuberculosis* infections in mice[9]. Interfering with bacterial biotin synthesis is therefore considered as a strategy to shorten TB chemotherapy. Enzymes involved in biotin synthesis have also been validated as targets for the development of antibiotics to treat infections with other bacterial pathogens such as *Francisella tularensis*, *Acinetobacter baumannii*, *Klebsiella pneumonia*, and *Pseudomonas aeruginosa*[10–12].

Biotin biosynthesis can be divided into the generation of the pimelate moiety and the assembly of the biotin rings. The enzymes synthesizing the pimelate moiety vary among biotin-producing

[1]Department of Microbiology and Immunology, Weill Cornell Medicine, New York, NY, USA. [2]Department of Medicinal Chemistry, University of Minnesota, Minneapolis, MN, USA. [3]Sloan Kettering Institute, Memorial Sloan Kettering Cancer Center, New York, NY, USA. [4]Department of Biochemistry and Biophysics, Texas A&M University, College Station, TX, USA. ✉e-mail: aldri015@umn.edu; dis2003@med.cornell.edu

organisms and consist of three BioH isoenzymes in *Mycobacterium smegmatis*[13,14]. Assembly of the biotin rings occurs via four conserved enzymes: BioF (7-keto-8-aminopelargonic acid (KAPA) synthase), BioA (7,8-diaminopelargonic acid (DAPA) synthase), BioD (dethiobiotin (DTB) synthetase), and BioB (biotin synthase)[15,16]. BioB, a member of the radical SAM superfamily[17], generates biotin by inserting a sulfur atom into DTB[18]. The activity of BioB from *E. coli* (BioB.ec) depends on an oxygen-sensitive $[4Fe-4S]^{2+}$ cluster and a $[2Fe-2S]$ cluster, which donates the sulfur that is incorporated into DTB[19,20]. This removal of a sulfur atom likely destroys the $[2Fe-2S]$ cluster and contributes to the low activity of purified BioB, which has been reported as less than one turnover per protein monomer in some studies[21,22]. Utilization of SAM, which is reductively cleaved by BioB, depends on the transfer of an electron from a donor protein. In *E. coli* this is mediated by flavodoxin (FldA) and flavodoxin (ferredoxin): NADPH oxidoreductase (Fpr)[21,23]; in mycobacteria, the electron donor system remains unknown. An elegant combination of bioinformatic, biochemical, and structural analyses recently demonstrated that the biotin synthases of most anaerobic bacteria differ from BioB.ec by utilizing a $[4Fe-5S]$ cluster instead of the $[2Fe-2S]$ cluster for biotin formation. Based on these studies biotin synthase can be classified into type I, exemplified by BioB.ec, and the type II enzymes present in anaerobic bacteria[24].

We previously reported that acidomycin, a natural product discovered in 1952[25–28], can prevent the growth of *M. tuberculosis* via inhibition of BioB[29]. However, we were unable to demonstrate inhibition of purified BioB from *M. tuberculosis* (BioB.tb) by acidomycin as we could not reconstitute active BioB.tb in vitro. Here we demonstrate that to be functional BioB.tb and BioB from *M. smegmatis* (BioB.sm) require small accessory proteins encoded by genes of previously unknown function, *rv1590* of *M. tuberculosis* and *msmeg3195* of *M. smegmatis*. This finding allowed us to reconstitute BioB.tb activity and demonstrate inhibition by acidomycin in vitro. Given their essentiality for the function of BioB.tb and BioB.sm we refer to Rv1590/Msmeg3195 as biotin synthase auxiliary proteins (BsaP).

## Results

### Mycobacterial genes are required to grow without extrabacterial biotin as identified by TnSeq

Transposon libraries of *M. tuberculosis* H37Rv and *M. smegmatis* were sequenced and compared after growth in media with and without biotin. As expected, transposon insertions in *bioA*, *bioF*, and *bioB*, which are required for the conversion of pimeloyl-ACP into biotin, were among the mutants most attenuated in biotin-free media (Supplementary Fig. 1). The screen was not sensitive enough to identify *bioD* as significantly attenuated. However, transposon insertions in the gene located directly downstream of *bioB* (*rv1590* and *msmeg3195*, here referred to as *bsaP*) attenuated *M. tuberculosis* and *M. smegmatis* in biotin-free media.

To confirm the essentiality of *bsaP* for growth without extrabacterial biotin, we deleted the predicted *bioB* operons, *bioB-bsaP-rv1591* and *bioB-bsaP-msmeg3196*, in *M. tuberculosis* and *M. smegmatis*, respectively. These mutants (Fig. 1a and Supplementary Fig. 2), Δ*bioBop*, were confirmed genotypically by whole-genome sequencing (Supplementary Fig. 2) and phenotypically by their failure to grow in biotin-free media (Fig. 1b and Supplementary Fig. 3a, b). Transformation with *bioB*op expression plasmids restored growth in biotin-free media of Δ*bioBop* in *M. tuberculosis* and *M. smegmatis*. Complementation also restored BioB expression in Δ*bioBop* to the level observed in *M. tuberculosis* H37Rv (Fig. 1c). Deletion mutants for each of the three *bioB* operon genes were constructed by transformation of the operon mutants with the appropriate expression plasmids (Fig. 1a). Δ*bioB* and Δ*bsaP* were unable to grow without extrabacterial biotin whereas deletion of the last gene in the bioB operon, *rv1591/msmeg3196*, had no impact on growth (Fig. 1d). We concluded that

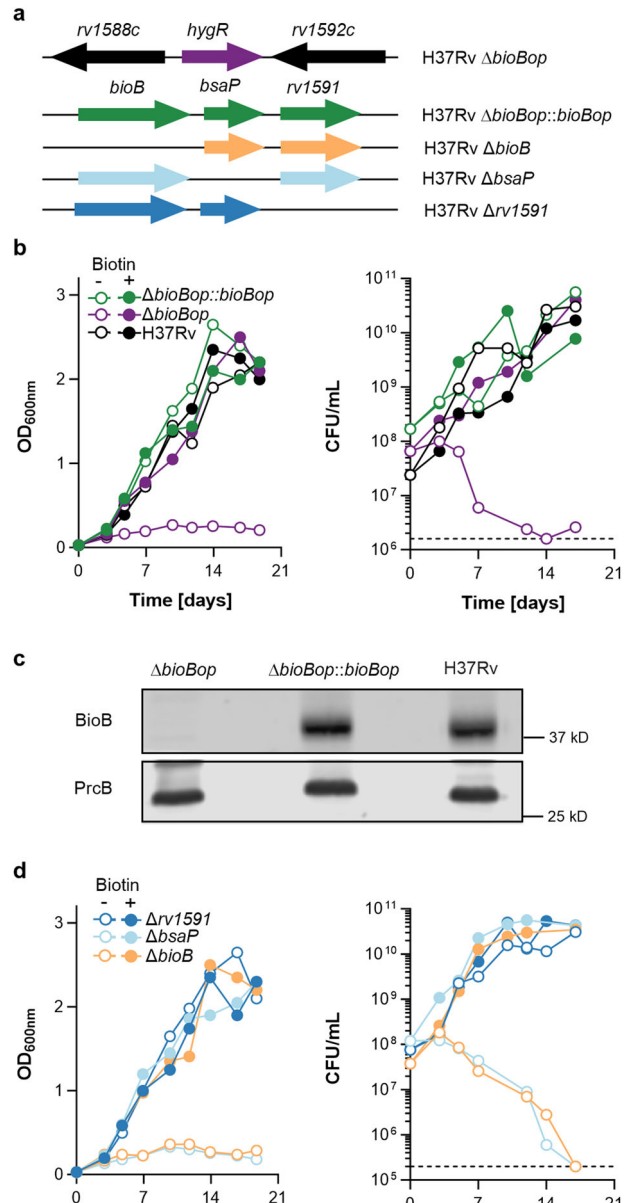

**Fig. 1 | *BsaP* is required by *M. tuberculosis* to grow without extrabacterial biotin. a** Mutant genotype. The first line displays the insertion of the hygromycin resistance marker (*hygR*) in Δ*bioBop*. The lines below specify the genes introduced by the transformation of Δ*bioBop* to generate single-gene deletion mutants. **b, d** Growth in modified 7H9 medium with (filled symbols) and without biotin (open symbols) as assessed by optical density measurements (left panels) and CFU enumerations (right panels). The dotted lines in the right panels of **b**, **d** specify the lower limit of detection. **c** Immunoblots for BioB and PrcB (loading control). The entire blot is shown in supplementary Fig. 11a. **c** Results represent at least three independent experiments. **b**, **d** Source data are provided as a Source data file.

*bsaP* is required by *M. tuberculosis* and *M. smegmatis* to grow in the absence of extrabacterial biotin.

### Chemical complementation of Δ*bsaP*

Chemical complementation was used to identify the reaction in the biotin synthesis pathway compromised by the deletion of *bsaP*. We first constructed *M. smegmatis* Δ*bioA-msmeg3196*, in which nine genes, *bioA* to *msmeg3196*, were replaced by a hygromycin resistance marker (*hygR*) (Supplementary Fig. 4a). Δ*bioA-msmeg3196* did not grow in biotin-free media and growth was restored after transformation with

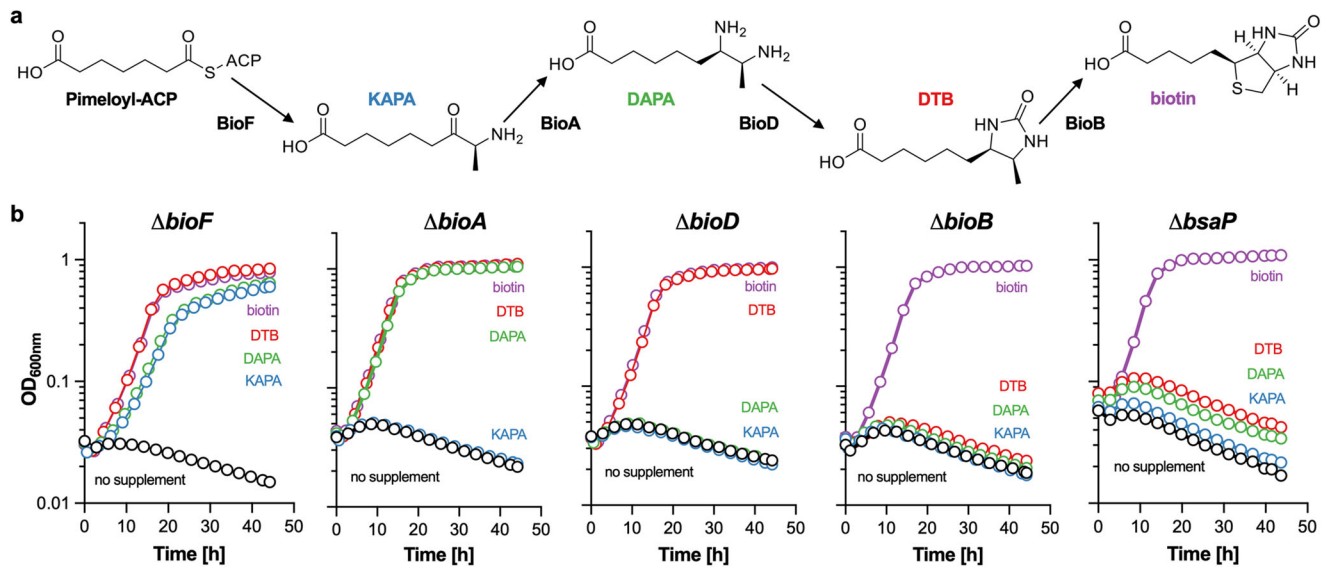

**Fig. 2 | Chemical complementation of *M. smegmatis* biotin auxotrophs. a** Biotin synthesis pathway. KAPA, DAPA, and DTB stand for 8-amino-7-oxononanoate, 7,8-diaminononanoate, and dethiobiotin, respectively. **b** Growth of *M. smegmatis* Δ*bioF*, Δ*bioA*, Δ*bioD*, Δ*bioB*, and Δ*bsaP* in modified biotin-free 7H9 medium with or without supplementation with KAPA, DAPA, DTB, or biotin. **b** Source data are provided as a Source data file.

an expression plasmid carrying the deleted genes (Supplementary Fig. 4b). *M. smegmatis* Δ*bioA-msmeg3196* was then used to generate single gene deletions for *bioF*, *bioA*, *bioD*, and *bioB*, each of which could be chemically complemented with the metabolite(s) generated downstream of the reaction compromised by the deleted gene (Fig. 2). For example, growth of *M. smegmatis* Δ*bioA* in biotin-free media was restored by DTB and DAPA, two metabolites generated downstream of the BioA reaction, but not by KAPA, the substrate of BioA. In contrast, *M. smegmatis* Δ*bioB* grew with biotin but not with KAPA, DAPA, or DTB. *M. smegmatis* Δ*bsaP* showed the same chemical complementation profile as *M. smegmatis* Δ*bioB* and only grew with biotin, indicating that *M. smegmatis* Δ*bsaP* was unable to convert DTB into biotin. We next transformed *M. smegmatis* Δ*bioBop* with expression plasmids for *bioB* and *bioB-bsaP* from *M. tuberculosis*. Analyses of the resulting strains demonstrated that *bioB* from *M. tuberculosis* only complemented the biotin-dependent growth defect of *M. smegmatis* Δ*bioBop* in conjunction with *bsaP* (Supplementary Fig. 3c). Taken together, these results suggested that the biotin synthases of *M. smegmatis* and *M. tuberculosis*, directly or indirectly, require *bsaP* to be functional.

## Impact of bsaP on biotin synthase expression and function in *M. smegmatis* and *Escherichia coli*

Immunoblots demonstrated that biotin synthase protein levels were not compromised in *M. smegmatis* Δ*bsaP* and *M. tuberculosis* Δ*bsaP* (Fig. 3a), which ruled out that *bsaP* is required for BioB expression. We next tested if the impact of *bsaP* on biotin synthase function depended on its native location, downstream of *bioB*. For this, we transformed *M. smegmatis* Δ*bioBop* with separate expression plasmids for *bioB* and *bsaP*. The *bioB* plasmid was integrated into the attachment site of the phage Tweety and the *bsaP* plasmid was integrated into the attachment site of the phage L5. The resulting strains, *M. smegmatis* Δ*bioBop::bioB.sm-Tw::bsaP.sm-L5* grew normally in biotin-free media (Fig. 3b). Similarly, *M. smegmatis bioBop* was also complemented when *bioB.tb* and *bsaP.tb* were expressed from different genomic locations (Supplementary Fig. 5a). Thus, the impact of *bsaP* on *bioB* did not depend on the location of *bsaP* within the genome.

The *E. coli* genome does not encode a homolog of *bsaP*, and *E. coli* biotin synthase can convert DTB to biotin in vitro without the need for an accessory protein. We, therefore, tested if the biotin synthase gene of *E. coli*, *bioB.ec*, was sufficient to complement Δ*bioBop* mutants of *M. smegmatis* and *M. tuberculosis*. Transformation with a *bioB.ec* expression plasmid resulted in *M. smegmatis* Δ*bioBop::bioB.ec* and *M. tuberculosis* Δ*bioBop::bioB.ec*, which grew without a need for extra-bacterial biotin (Fig. 3b and Supplementary Fig. 5b). This confirmed that the sole reason for the inability of *M. smegmatis* Δ*bioBop* and *M. tuberculosis* Δ*bioBop* to grow in biotin-free media was a lack of biotin synthase activity.

Next, we transformed *E. coli* Δ*bioB* with expression plasmids carrying either only *bioB* from *M. smegmatis* or *M. tuberculosis*, or the mycobacterial *bioB* together with *bsaP*. Complementation of *E. coli* Δ*bioB* with bioB from either mycobacterial species depended on *bsaP* (Fig. 3c). Again, *bsaP* had no impact on BioB expression (Fig. 3d). From these experiments we concluded that biotin synthases from *M. smegmatis* and *M. tuberculosis* require *bsaP* to be functional.

## Importance of BsaP for the activity of purified BioB.tb

Previously, we were unable to reconstitute the activity of BioB.tb in vitro, a result that we hypothesized was due to the failure of the heterologous flavodoxin system to activate BioB.tb[29]. Upon the discovery of BsaP, we sought to examine if BsaP could restore the function of BioB.tb in vitro. Consistent with the genetic complementation studies, BsaP promoted the conversion of DTB to biotin by BioB.tb in vitro under anaerobic conditions (Fig. 4a). No difference in biotin production was observed when BsaP was added to BioB.ec. The reaction progress curve shows addition of BsaP to BioB.tb led to biotin production immediately without a distinguishable lag period (Fig. 4b). Although, we were unable to attain steady-state turnover, a fit of the initial velocity vs DTB concentration at fixed saturating concentrations of SAM at 25 °C provided a dissociation constant for DTB of 7 μM presuming rapid equilibrium and a maximal rate of $0.0064 \, min^{-1}$ (Fig. 4c), which is similar to the data obtained with BioB.ec[30]. By varying the ratio of BsaP vs BioB.tb, we showed the maximal rate is saturable and fitting of the initial velocity to the ratio of BioB.tb−BsaP provided the dissociation equilibrium constant ($K_D$) of 0.95 μM for BioB.tb−BsaP complex formation (Fig. 4d).

We previously demonstrated that the natural product acidomycin targets BioB[29], but were unable to biochemically validate these findings since our assay lacked BsaP. We re-examined acidomycin in our improved BioB.tb−BsaP biochemical assay, which revealed that acidomycin is indeed a potent inhibitor of BioB.tb displaying an inhibition

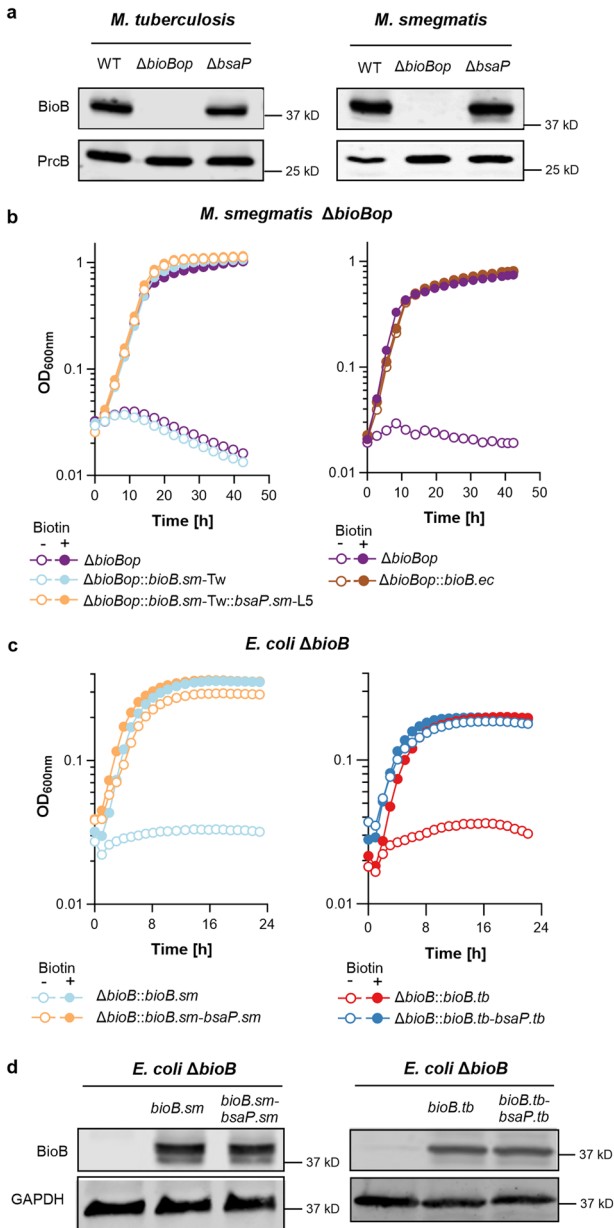

**Fig. 3 | Impact of *bsaP* on growth of *bioB* mutants and BioB expression.**
**a** Immunoblots for expression of BioB and PrcB (loading control) in *M. tuberculosis* and *M. smegmatis*. The entire blot is shown in Supplementary Fig. 11b.
**b**, **c** Growth of *M. smegmatis* Δ*bioBop* (**b**) and *E. coli* Δ*bioB* (**c**) with and without extrabacterial biotin. **d** Immunoblots for expression of BioB and GAPDH (loading control) in *E. coli*. The entire blot is shown in Supplementary Fig. 11c. **a**, **d** Results represent at least three independent experiments. **b**, **c** Source data are provided as a Source data file.

constant $K_i$ of = 0.11 μM (Supplementary Fig. 6). These experiments validated the prediction that BsaP facilitates the conversion of DTB into biotin and established and in vitro assay, which can be used to identify and characterize inhibitors of mycobacterial biotin synthases.

## BsaP structure predictions, Fe–S cluster formation, and phylogenetic analyses
The three-dimensional structures of BsaP or its homologs have not been determined experimentally. However, alphafold (AF) predicts almost identical structures for BsaP.tb and BsaP.sm with high confidence scores (Fig. 5a). The predicted structures are unique, showing no homology to any known structures when searched by VAST[31]

against the pdb database. Interestingly, although BsaP lacks the conserved motif CXX-C/A/G/H-XXC(X)ₙCP for known Fe–S clusters [4Fe–4S]⁺ and [3Fe–4S]⁺[32], AF predicts three conserved cysteines-Cys44, Cys47, and Cys64 of BsaP.tb and Cys45, Cys48, and Cys65 of BsaP.sm-to be positioned in a way that would allow the formation of a Fe–S cluster. While neither the tertiary structure or sequence motif spanning cysteines are consistent with any described ferredoxins, the constellation of cysteines and their surrounding amino acids is similar to known [3Fe–4S]⁺ ferredoxins, such as *Rhodopseudomonas palustris* HaA2 (4ID8, Fig. 5a).

To define the importance of the conserved cysteines we mutated Cys44, Cys47, and Cys64 of BsaP.tb individually or collectively into alanines. The mutated proteins were unstable, which prevented us from determining if these cysteines are required for the function of BsaP.tb (Supplementary Fig. 7). Next, we analyzed the formation of a possible iron-sulfur cluster by UV spectroscopy. Peaks at 320 nm and 452 nm are associated with [2Fe–2S]⁺ clusters, while a shoulder at 420 nm can be due to a combination of a weak band associated with a [2Fe–2S]⁺ cluster and stronger bands associated with [4Fe–4S]⁺ or [3Fe–4S]⁺ clusters bound in the radical SAM cluster-binding sites[30]. Anaerobic reconstitution of BsaP resulted in an increase of absorbance between 420 nm and 452 nm compared to BsaP prepared under aerobic conditions (Fig. 5b), which suggests that BsaP may form an oxygen-sensitive Fe–S cluster. Spectral analyses of BioB.tb indicates that it's [2Fe–2S]⁺ and [4Fe–4S]⁺ clusters that can form independently of BsaP even though the conversion of DTB to biotin depended on reconstituted BsaP. Minimal biotin synthesis was observed when BioB.tb was combined with BsaP that had not been reconstituted anaerobically. The combination of independently reconstituted BioB.tb and BsaP elevated the yield of biotin to an extent, but only when both BioB.tb and BsaP were reconstituted together with Fe³⁺ and Na₂S did BioB.tb obtain maximal observed activity (Fig. 5c).

Homology searches identified hundreds of BsaP homologs from various actinobacteria, with representative sequences shown in Supplementary Fig. 8a. Most of the encoding genes are located downstream of the respective *bioB* (Supplementary Fig. 8b). Inspection of a consensus maximum log-likelihood phylogenetic tree of 41 BsaP homologs identified the three cysteines we postulate to form an atypical Fe–S cluster to be conserved and located in the most conserved region of the BsaP alignment (Fig. 5d). Taken together, these data suggest that formation of a Fe–S cluster by BsaP is likely required to achieve biotin synthesis by BioB.tb.

## Structural analyses of BsaP-associated and BsaP-independent biotin synthases
We used the presence or absence of BsaP homologs in a genome to categorize biotin synthases as BsaP-associated or BsaP-independent. An alignment of their amino acid sequences revealed that the primary structures of BsaP-associated biotin synthases, which are common in actinobacteria, cluster together and are more similar to each other than to their BsaP-independent counterparts (Supplementary Fig. 9).

Next, we tried to understand the structural basis of BsaP's function by comparing the BioB.ec X-ray structure with AF structure predictions for BioB.tb and BioB.sm. All three proteins fold into similar tertiary structures (Supplementary Fig. 10). The BioB.ec amino acids that form the [4Fe–4S]⁺ cluster-Cys53, Cys47, and Cys60-are conserved in BioB.tb and BioB.sm (Fig. 6a) along with other cluster surrounding residues, suggesting that this cluster functions similarly in BioB.ec and BioB.tb/sm. In contrast, the residues forming the [2Fe–2S]⁺ cluster show striking differences between BioB.ec and BioB.tb/sm (Fig. 6b). In BioB.ec the [2Fe–2S]⁺ is coordinated by Cys97, Cys128, Cys188, and Arg260. In BioB.tb and BioB.sm Cys128 is replaced by Ala160 and Ala164, respectively. Other notable differences include symmetrical replacements of two serine residues of BioB.ec (Ser43 and Ser218) with glutamates (Glu75 and Glu248 in BioB.tb), Tyr149 with

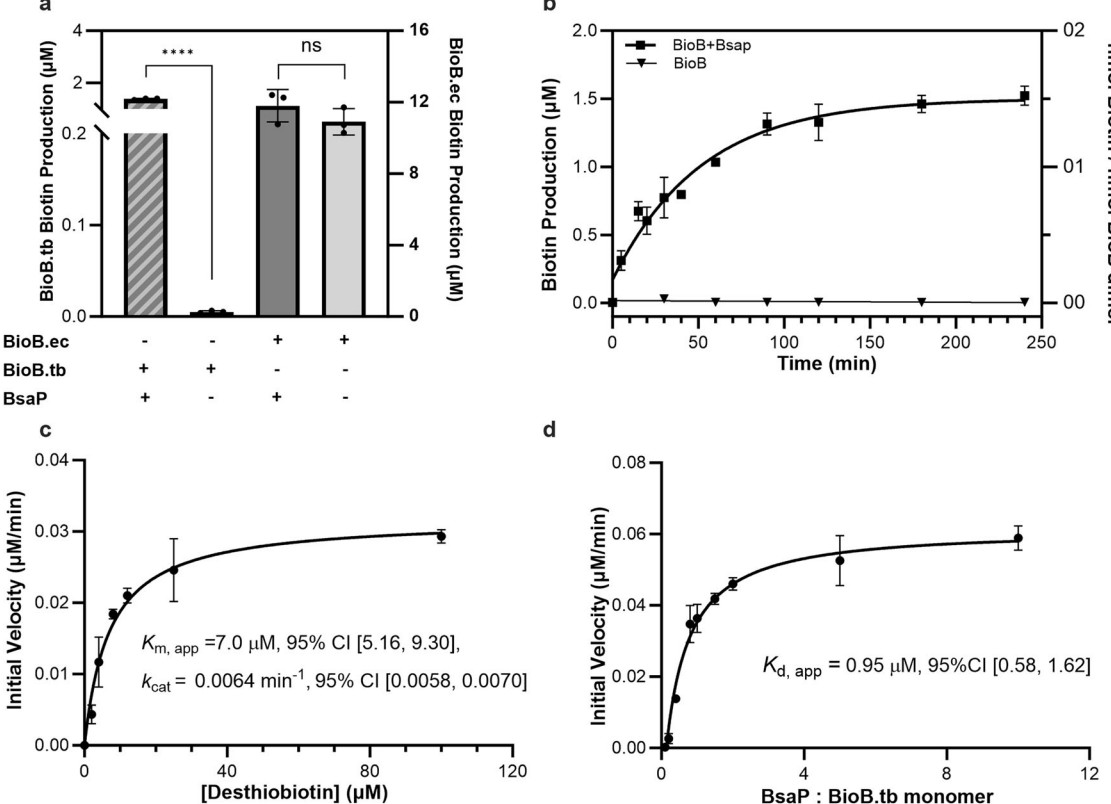

**Fig. 4 | Biochemical characterization. a** BsaP is essential for the biotin synthesis catalyzed by BioB.tb, but not BioB.ec. **b** Progress curve of biotin synthesis catalyzed by BioB.tb (5 μM) with and without Bsap (5 μM). **c** Saturation curve of $v_0$ vs [DTB]. The assay was performed under initial velocity conditions with a fixed saturating concentration of SAM (100 μM) while varying [DTB]. The $K_M$ and $k_{cat}$ values were measured by fitting the $v_0$ vs [DTB] to Eq. 1. **d** Saturation curve of BioB.tb + BsaP

showing the effect of increasing BsaP concentration on the overall rate of turnover. The $K_D$ of BioB.tb and BsaP interaction was measured by fitting the $v_0$ vs [BsaP] to Eq. 2. **a**–**d** $n = 3$. Two-sided unpaired $t$ test; ****$p = 1.3537E{-}07$, $p = 0.2520$ (ns); ns $p > 0.05$, *$p < 0.05$, **$p < 0.01$, ***$p < 0.001$ (**a**). Error bars, mean ± SD. Source data are provided as a Source data file.

Arg181, and Arg95 with Glu126. These marked differences in residues surrounding the [2Fe–2S]$^+$ cluster suggest that this cluster functions differently in BioB.tb/sm than BioB.ec and may not be fully functional on its own in BioB.tb/sm. Interestingly, these replacements could be seen across aligned BsaP-dependent BioB sequences (Supplementary Fig. 9).

Sequence analyses also detected a BioB–BsaP fusion protein in genomes of *Pseudonocardia* (Fig. 6c). The AF prediction for this protein places the BsaP-like domain in the vicinity of the biotin synthase [2Fe–2S]$^+$ cluster, further suggesting that BsaP might be required to perform the [2Fe–2S]$^+$-mediated insertion of a sulfur atom into DTB. Electrostatic surface analysis of BioB.tb identified a negatively charged patch close to the [2Fe–2S]$^+$ cluster access, while the surface of BsaP, closest to the cysteins and potential Fe–S cluster, has a positive charge, mainly contributed by Arg49, Arg63, and Arg66 (all highly conserved among homologs), making these surfaces likely candidates for protein–protein binding between BioB and BsaP (Fig. 6d). Unfortunately, a BsaP.tb mutant in which two arginines mainly contributing to the positively charged surface patch was replaced with glutamate (R49E, R66E) was also unstable (Supplementary Fig. 10), which again prevented us from determining the importance of these residues for function. Nevertheless, these structural and sequence analyses suggest that biotin synthases that depend on BsaP require this accessory protein to support function distinct from BioB.ec [2Fe–2S]$^+$ cluster. They further predict that the conserved cysteine residues of BsaP directly interact with a Fe–S cluster that is required for function and identify residues in BsaP that mediate a direct interaction with BioB.

## Discussion

This work discovered a novel class of auxiliary proteins, which we have named BsaP, that are required for the activity of mycobacterial biotin synthases in vivo and in vitro. Deletion of the encoding gene, *bsaP*, causes biotin auxotrophy in *M. smegmatis* and *M. tuberculosis*. The growth defect *bsaP* deletion mutants exhibited in biotin-free media can be complemented with biotin but not with another intermediate of the biotin synthesis pathway, which associates the function of BsaP to the conversion of DTB into biotin. The prediction that BsaP is required for this last step in biotin synthesis was confirmed by the following lines of evidence: (i) *BsaP* deletion mutants can be complemented by BioB.ec. (ii) *BioB*.tb and *bioB*.sm requires *bsaP* to complement *E. coli* Δ*bioB*. (iii) Purified BioB.tb is inactive in vitro, but can convert DTB into biotin when provided with BsaP. Taken together, these data demonstrate that the biotin synthases of *M. smegmatis* and *M. tuberculosis* require BsaP as an accessory factor to be active. The widespread distribution of BsaP homologs in actinobacteria suggests that the activity of most actinobacterial biotin synthases may be dependent on BsaP-like small accessory proteins.

The poor activity of BioB.tb we measured is reminiscent of the first in vitro reconstitutions of BioB.ec, which were notoriously difficult with initial reports yielding less than one turnover per BioB dimer due to complications resulting from protein inactivation[30,33,34]. It seems likely that biotin production by BioB.tb–BsaP may not be limited to this rate in vivo. Importantly, the current biochemical BioB.tb–BsaP assay was sufficient to demonstrate the susceptibility of BioB.tb to inhibition by acidomycin, which shows its potential to identify small-molecule inhibitors. The inhibitory constant was measured for BioB.tb

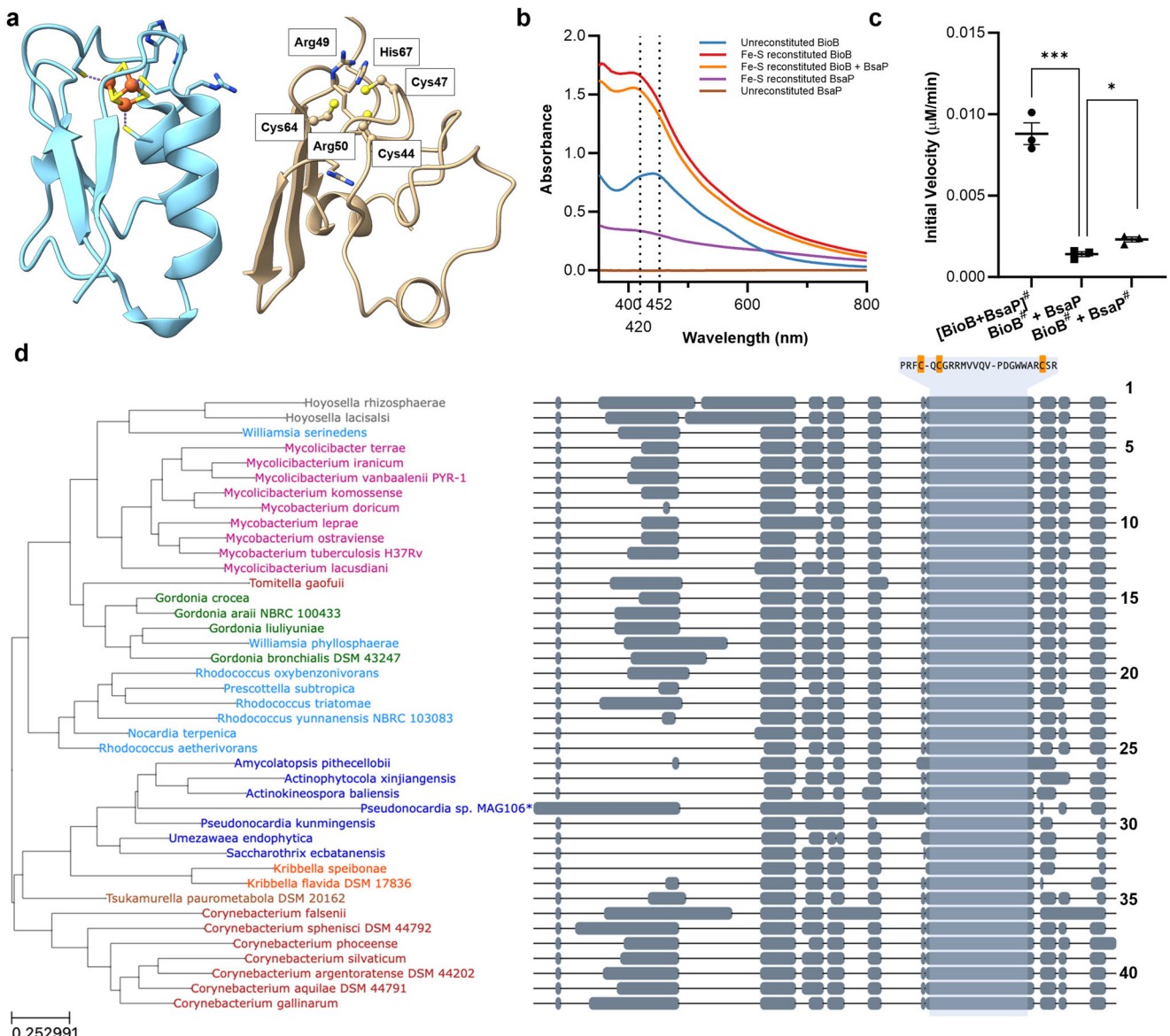

**Fig. 5 | BsaP structure prediction, Fe−S cluster formation, and phylogenetic analyses. a** Comparison of the crystal structure of a [3Fe−4S] + ferredoxin (4ID8, left) with the AF prediction for BsaP.tb (right). Impact on Fe−S cluster reconstitution on the UV-spectra of BioB.tb and BsaP.tb. The dashed lines represent 420 nm and 452 nm, respectively (**b**) and the enzymatic activity of BioB.tb (**c**). #Indicates Fe−S cluster reconstitution with Fe (II/III) ammonium sulfate and Na₂S. [BioB + BasP]* was reconstituted as a mixture of both proteins. **d** Consensus maximum log-likelihood phylogenetic tree of BsaP homologs. To the right of the tree, a CDS multiple sequence alignment for the BsaP homologs is shown, where aligned portions are drawn in gray, and alignment gaps are depicted by a line. The BsaP consensus sequence is shown for the region containing the conserved cysteine residues (Cys44, Cys47, and Cys64 of BsaP.tb), which are highlighted in orange. **c** $n = 3$. Two-sided unpaired $t$ test; ***$p = 0.0004$, **$p = 0.0141$; ns $p > 0.05$, *$p < 0.05$, **$p < 0.01$, ***$p < 0.001$. Error bars, mean ± SD. **b, c** Source data are provided as a Source data file.

($K_i = 0.27 \, \mu M$) is similar to that for BioB.ec ($K_i \sim 1 \, \mu M$)[29]. This agrees with our previous work suggesting that the species-selectivity of acidomycin is likely due to selective intrabacterial accumulation[29].

The exact mechanism by which BsaP assists mycobacterial biotin synthases remains to be determined. However, our analyses clearly point towards the BioB [2Fe−2S]⁺ cluster-dependent transfer of a sulfur atom into DTB as a BsaP-dependent step. BsaP forms an unusual Fe−S cluster that participates in this reaction. After we identified BsaP as a Fe−S-containing protein, an independent work reported BsaP (Rv1590) as a mycobacterial rubredoxin-like protein capable of binding iron and showing redox activity. Although the importance of BsaP for biotin synthesis was not demonstrated, their general results are consistent with our findings[35]. Our studies further suggest that the BsaP-dependent biotin synthases utilize a biochemical mechanism that is distinct from both type I (e.g., BioB.ec) and recently identified type II biotin synthases[24]. In-depth characterization of BsaP may also help to better understand other radical SAM enzymes, such as the lipoyl synthase LipA. LipA catalyzes the insertion of sulfur during the last step of lipoic acid biosynthesis and may also be aided by a BsaP-like accessory protein. However, mycobacterial LipA does not require BsaP to be active in vitro[36]. Furthermore, mycobacteria require LipA to grow in standard media, while BsaP is dispensable for growth under these same conditions. Whether BsaP or similar accessory proteins can enhance LipA activity or assist other Fe−S cluster enzymes thus remains to be determined. The discovery of this family of redox-active proteins will enable further mechanistic studies on biotin synthases and may help to initiate new antitubercular drug discovery projects.

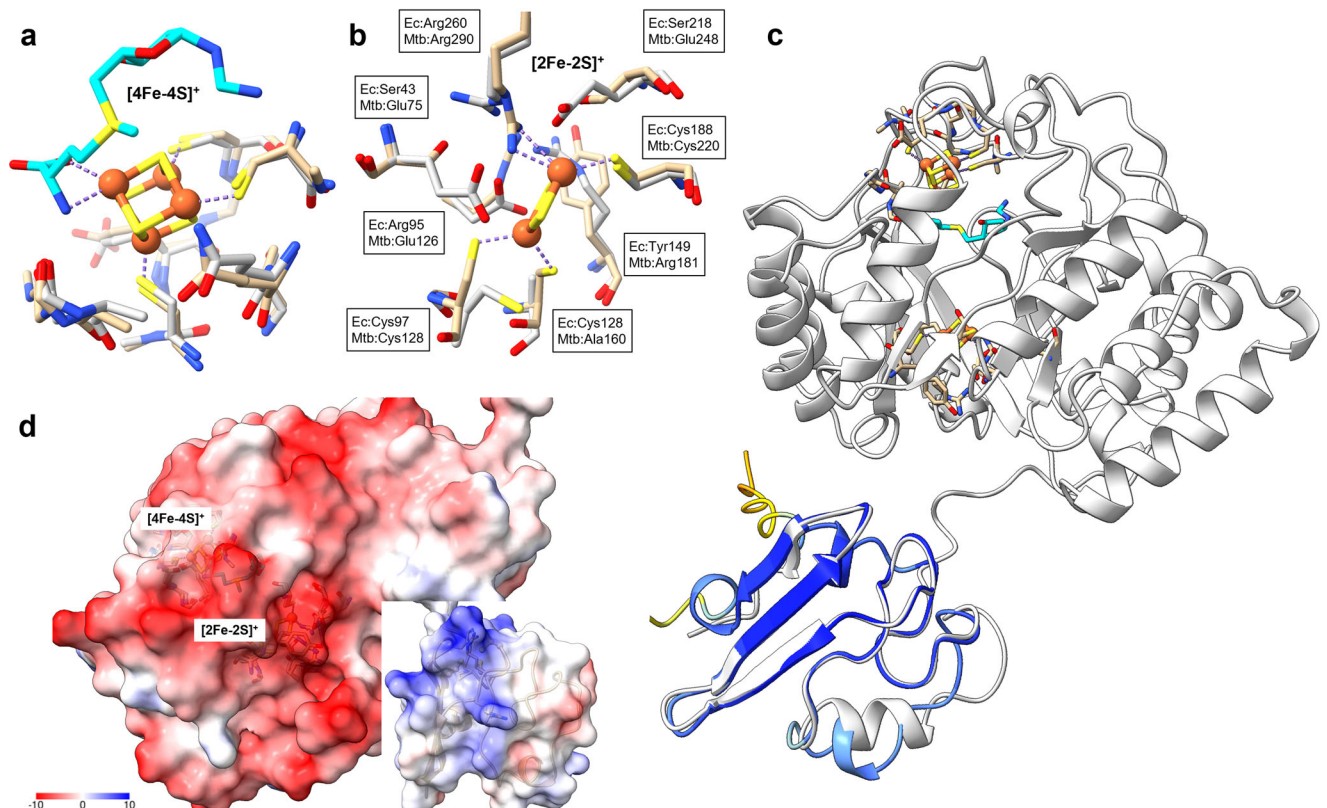

**Fig. 6 | Structural analyses.** Comparisons of the amino acids forming the [4Fe–4S]+ (**a**) and [2Fe–2S]+ (**b**) clusters of BioB.ec (1R30, protein carbons are shown in tan carbons, SAM carbons are shown in teal), and BioB.tb (AF prediction, gray carbons). **c** Alignment of the AF predictions for BsaP.tb (colored by AF confidence score) and AF predicted structure of BioB from Pseudonocardia (light gray) with an overlay of Fe–S clusters and substrates from the BioB.ec crystal structure (tan). **d** Electrostatic surface analyses of the AF predictions for BioB.tb (left) and BsaP.tb (right).

## Methods

### Bacterial strains and culture

*M. tuberculosis* H37Rv and *M. smegmatis* mc²-155 were cultured in Middlebrook 7H9 media supplemented with 0.2% glycerol, 0.02% tyloxapol, and 10% ADNaCl (0.5% fatty acid-free BSA, 0.2% dextrose, and 0.085% NaCl). *M. tuberculosis* and *M. smegmatis* mutants were cultured in a modified 7H9 medium (Supplementary dataset 1) supplemented with 0.2% glycerol, 0.02% tyloxapol, 10% ADNaCl, and appropriate antibiotics. Unless otherwise indicated, *M. tuberculosis* and *M. smegmatis* mutants were cultured in a modified 7H9 medium plus or minus 1 μM biotin. *E. coli* JW0758 *ΔbioB* was cultured with minimal M9 medium (Supplementary dataset 1) supplemented with 0.1 U/mL avidin or 1 μM biotin and appropriate antibiotics. *M. tuberculosis* was grown standing in tissue culture flasks; *M. smegmatis* and *E. coli* were grown in Erlenmeyer flasks or culture tubes with shaking. All cultures were grown and maintained at 37 °C.

### Mutant construction

We obtained *M. smegmatis ΔbioA-msmeg3196, ΔbioB-bsaP-msmeg3196*, and *M. tuberculosis ΔbioB-bsaP-rv1591* deletion mutants through recombineering by using *M. smegmatis* mc²-155 and *M. tuberculosis* H37Rv expressing the recombinase RecET[37]. DNA fragments with the hygromycin resistance gene (*hygR*) flanked by 500 bp upstream and downstream of the target loci were obtained from GeneScript). Strains and plasmids are listed in Supplementary datasets 2 and 3, respectively.

### Whole genome sequencing

The genetic identity of *Msm ΔbioA-msmeg3196, ΔbioB-bsaP-msmeg3196*, and *M. tuberculosis ΔbioB-bsaP-rv1591* was confirmed by whole genome sequencing (accession numbers SAMN30984225,

SAMN30984226, and SAMN30984221). Genomic DNA was isolated, sequenced, and analyzed as in our previous work[29,38,39].

### Construction of Tn libraries

*M. tuberculosis* H37Rv and *M. smegmatis* mc²-155 transposon libraries were constructed by Himar1 mutagenesis as previously described[40]. Briefly, 100 mL of mid-log phase culture (optical density at 600 nm, ~0.7–1.0) was incubated with approximately 2 × 10¹¹ PFU of ΦMycoMarT7 phage[41] at 37 °C for 4 h. The cultures were then washed, spread on agar plates with or without biotin, and incubated for 18 days or 3 days for *M. tuberculosis* or *M. smegmatis*, respectively, at 37 °C. Libraries with titers of greater than 100,000 CFU (colony forming unit) were used in the analysis to ensure comprehensive coverage of the possible TA dinucleotide insertion sites in the mycobacterial genome.

### Sequencing of Tn libraries

Genomic DNA was isolated from selected transposon libraries, and the library mutant composition was determined by sequencing amplicons of the transposon-genome junctions as previously described[40]. On average, library sequencing yielded between 1.1 million and 2.7 million unique transposon-genome junction reads, covering between 54 and 66% of the possible TA dinucleotide insertion sites in the genome for *M. tuberculosis* and between 0.7 million and 1.2 million unique transposon-genome junction reads, covering between 42 and 72% for M. *smegmatis* (NCBI accession numbers SRR22070226 to SRR22070233 for *M. tuberculosis* and SRR22069860 to SRR22069863 for *M. smegmatis*).

### Data processing

Raw sequence data were processed using the TPP tool from the TRANSIT TnSeq analysis platform[42], and transposon genome junctions

were mapped to the *M. tuberculosis* H37Rv or *M. smegmatis* mc$^2$-155 reference genome (NC_018143.1, NC_008596.1, respectively) using the Burroughs-Wheeler aligner (BWA). The final template counts were normalized across all the data sets using Trimmed Total Reads (TTR) normalization. TTR normalizes data sets so that they have the same mean template count while ignoring ("trimming") the top and bottom 5% of reading counts to reduce the influence of outliers.

### Immunoblots

*M. tuberculosis* cell lysates were prepared by bead-beating cell pellets in PBS-PI (phosphate-buffered saline buffer-protease inhibitor) and sterilized by passage through a 0.22 mm Spin-X filter (Costar Spin-X, 8160). *M. smegmatis* lysates were prepared by bead-beating cell pellets in PBS-PI (phosphate-buffered saline buffer-protease inhibitor). *E. coli* cell lysates were prepared in CelLytic™ B 10× lysis buffer (Sigma, SLBW8423). For immunoblotting, lysates were denatured by heating at 95 °C for 15 min. Standard SDS-polyacrylamide gel electrophoresis was performed, and proteins were transferred onto a nitrocellulose membrane(Invitrogen, IB23001). Membranes were blocked with intercept (PBS) blocking buffer (LI-COR, 927-70001) for 30 min at room temperature. Rabbit anti-BioB.tb, rabbit anti-BioB.sm, rabbit anti-PrcB, rabbit anti-DlaT, mouse anti-HA, and mouse anti-GAPDH were used as the primary antibodies at a dilution of 1:1000 overnight at 4 °C. Goat anti-mouse IgG or goat anti-rabbit IgG (LI-COR Biosciences) was used as the secondary antibody at a dilution of 1:0000 at 30 min at room temperature. Blots were developed using the Odyssey Infrared Imaging System (LICOR Biosciences). The signal intensities of the bands were analyzed in ImageJ. Immunoblotting antibodies: rabbit anti-BioB.tb (GenScript, Order ID: U8595BJ180-1, lot no.: A317030069), rabbit anti-BioB.sm (GenScript, Order ID: U7648CC310-7, lot no.: A317070075), rabbit anti-PrcB (GN-13.782), rabbit anti-DlaT (We thank Ruslana Bryk and Carl Nathan for DlaT-specific antiserum), mouse anti-GAPDH (Abcam, lot no.: ab125247), and mouse anti-HA (ThermoFisher 26183, lot no.: TJ272546). Full western blot images are shown in Supplementary Fig. 11.

### Growth and survival of *M. tuberculosis*

Bacteria were grown to mid-log phase in 7H9 media, washed twice in modified 7H9, and inoculated into modified 7H9 media plus or minus 1 μM biotin. The initial inoculum was $1.67 \times 10^7$ CFU/mL (OD580nm: 0.03). Growth was monitored by measuring the optical density (OD580nm) over the time course of the experiment. To determine the survival (CFU/mL) of the bacteria, serial dilutions of the cultures were made at indicated time points in PBS and plated onto 7H11 solid agar plates (without antibiotics) containing glycerol and supplemented with OADC followed by incubation for 3 weeks at 37 °C.

### Growth and chemical complementation of *M. smegmatis*

Bacteria were grown to mid-log phase in 7H9 media, washed twice in modified 7H9, and inoculated into modified 7H9 media plus or minus 1 μM biotin. For chemical complementation experiments, mid-log phase bacteria were washed twice in modified 7H9 and then inoculated into modified 7H9 media supplemented with 10 μM KAPA, 10 μM DAPA, 1 μM DTB, or 1 μM biotin, as well as no supplement control. Growth was monitored by measuring the optical density at 580 nm.

### Statistical analysis

All experiments were performed as biologically independent experiments with technical replicates, and at least two biologically independent experiments were performed. GraphPad Prism 9.3.1 (GraphPad Software) was used to perform statistical tests and generate graphs. The statistical tests used are indicated within figure legends and the corresponding *p* values are reported as *$p < 0.05$, **$p < 0.01$, ***$p < 0.001$, and ****$p < 0.0001$.

### Expression and purification of recombinant proteins

Plasmids used for heterologous protein expression in *E. coli* BL21(DE3) are listed in Supplementary dataset 2. For the production of flavodoxin (FldA), ferredoxin:NADP$^+$ oxidoreductase (FNR), and MTA/SAH/5′-dAH nucleosidase (MtnN), *E. coli* carrying the targeted recombinant plasmid (Supplementary dataset 2) was inoculated into 10 mL of LB medium containing 50 μg/mL kanamycin and shaken at 37 °C, 250 rpm overnight. Next, 1 L of LB media with 50 μg/mL kanamycin was inoculated with the overnight culture and incubated at 37 °C with shaking at 200 rpm until the OD$_{600}$ reached 0.6–0.8. IPTG (Invitrogen, cat. 15529019) was added to a final concentration of 100 μM. For *E. coli* harboring the FldA plasmid, riboflavin (Chem-Impex) was also added to the culture to a final concentration of 5 mg/L. The cell culture was incubated at 18 °C for 16 h. On the next day, bacteria were harvested by centrifugation at 4000×g for 20 min at 4 °C and frozen at −80 °C until usage. To produce BioB, 5 mL overnight culture of *E. coli* strains containing BioB.tb or BioB.ec plasmids were inoculated into 1 L of LB media containing 50 μM iron (III) citrate (Sigma Aldrich) and 50 μg/mL kanamycin. The culture was incubated at 37 °C with shaking at 250 rpm until OD$_{600}$ 1.0–1.5. The cell culture was then cooled to 25 °C with shaking at 250 rpm for 30 min. After adding IPTG to a final concentration of 100 μM, the cell culture was incubated for 3–5 h at 25 °C with slower shaking at 100 rpm. The cells were pelleted by centrifugation after cooling on ice for 30 min. Cells were resuspended and washed with cold 50 mM Tris, pH 8.0 once before storage at −80 °C. After induction with 300 μM IPTG, *E. coli* cells harboring the BsaP gene were incubated at 37 °C for 5 h and then pelleted by centrifugation and stored at −80 °C for later usage.

Frozen cells were thawed on ice and resuspended in ice-cold lysis buffer (50 mM Tris, pH 8.0, 1 mg/mL lysozyme, Roche cOmplete™ inhibitor cocktail, 10 U DNase). The suspended cells were lysed using a French Press. The histidine-tagged proteins were separated from cell lysate using HisTrap HP 5 mL column (Cytiva, cat. 17-5248) operated with NGC Quest 10 Plus Chromatography system (Bio-Rad, cat. 7880003). The column was equilibrated with buffer A (50 mM Tris, pH 8.0, 500 mM NaCl, and 20 mM imidazole) prior to applying the cell lysate. The column was then washed with ten CVs of buffer A. The bound protein was eluted with an increasing percentage of buffer B (50 mM Tris, pH 8.0, 500 mM NaCl, 500 mM imidazole). GST-tagged BsaP was purified using a GSTrap FF column (Cytiva, cat. 17-5130) following the manufacturer's instructions. After the affinity purification step, proteins were purified further with gel filtration chromatography using HiLoad 16/600 Superdex 75 pg (GE Healthcare, cat. 28-9893-33).

### Quantification of recombinant proteins

All estimations for protein quantification were conducted on Nano-Drop One (Thermo Fisher). Unless specified, protein concentrations were estimated using extinction coefficient at 280 nm. The concentrations of FldA and FNR were estimated using $\varepsilon_{466} = 8250$ M$^{-1}$ cm$^{-1}$ and $\varepsilon_{468} = 7240$ M$^{-1}$ cm$^{-1}$, respectively. The concentrations of BioB.tb and BioB.ec were estimated using $\varepsilon_{452} = 8400$ M$^{-1}$ cm$^{-1}$. The A$_{420}$/A$_{452}$ ratio of the fully reconstituted BioB provides information on the ratios of the two Fe–S clusters[43]. The reported ratio for *Ec*BioB is 1.18 which corresponds to 1:1 ratio of [2Fe−2S]$^{2+}$ and [4Fe−4S]$^{2+}$. Anaerobic reconstitution of *Mt*BioB+Bsap gave an A$_{420}$/A4$_{52}$ (1.53/1.31) of 1.17.

### BioB activity assay

All experiments were performed with triplicates except for the time-dependent enzyme kinetics study that was performed in duplicate. All procedures and incubations were performed in an anaerobic chamber (Mbraun Glovebox) with an O$_2$ level < 5 ppm, following the procedure published by Jarret and colleagues[43] with modifications. Briefly, BioB.tb or BioB.ec proteins in 50 mM Tris-HCl, pH 8.0, and 10 mM NaCl were prepared to provide a final concentration of 5 μM and a final reaction volume of 200 μL after the addition of all assay components.

For assays with BsaP, the protein was added to the BioB protein solution at the indicated concentration. To reconstitute active $[4Fe-4S]^{2+}$ cluster in BioB, 5 mM DTT, 200 μM $Na_2S$, and 200 μM ferrous ammonium sulfate/ferric ammonium sulfate were added as final concentrations in order to the solution and mixed well. The mixture was incubated at room temperature for 10 min. Next, FldA (25 μM), FNR (5 μM), NADPH (1 mM, Thermo Fisher Scientific, cat. 328742500), DTB (100 μM, Sigma Aldrich, cat. D1411), and MtnN (70 nM) were added to the reconstituted BioB enzyme solution (final concentrations given in parenthesis). S-adenosyl methionine (100 μM, Carbosynth) was added to initiate the reaction, which was then incubated at 37 °C for the indicated time. The reaction was quenched with 20 μL acetic acid/ sodium acetate, pH 4.5. For the determination of kinetic parameters for DTB, the reaction was measured under initial velocity conditions (10 min) using the fixed saturating concentration of DTB (100 μM) and SAM (100 μM) and a two-fold dilution series of DTB (25–1.56 μM and 100 μM). The substrate saturation kinetic data were fit by nonlinear regression analysis to the Michaelis–Menten equation (Eq. 1) using Prism 7.0. For determination of the $K_D$ of the BioB and BsaP association, the reaction was measured under initial velocity conditions (10 min) using the fixed saturating concentration of DTB (100 μM) and SAM (100 μM) while varying BsaP (50–0.2 μM) using a two-fold dilution series. The BsaP saturation kinetic data were fit by nonlinear regression analysis to Eq. 2 using Prism 7.0 following the analysis described by Khosla[44]. For inhibition studies with acidomycin, a 1.5-fold dilution series of acidomycin (45–0.34 μM) was added 10 min prior to initiation of the reaction. The reaction was performed under initial velocity conditions (10 min) using fixed saturating concentrations of DTB (10 μM) and SAM (100 μM). The normalized initial velocities ($v_i/v_0$) were fit by nonlinear regression analysis to the Morrison equation (Eq. 3) using Prism 7.0 to determine the apparent $K_i$ value.

$$v_0 = \frac{V_{max}[S]}{K_M + [S]} \tag{1}$$

$$v_0 = \frac{V_{max}[Bsap]}{K_D + [Bsap]} \tag{2}$$

$$\frac{v_i}{v_0} = 1 - \frac{([E] + [I] + K_i^{app}) - \sqrt{([E] + [I] + K_i^{app})^2 - 4[E][I]}}{2[E]} \tag{3}$$

### Biotin quantification in assay samples

Biotin levels in samples were quantified by LC-MS. Prior to analysis on LC-MS, samples were centrifuged at 18,000×g for 15 min to remove precipitated proteins and passed through a 0.2 μm PVDF filter (Millipore Sigma). Next, 0.5 μM Biotin-$d_4$ (Cambridge Isotope Laboratories, cat. DLM-9751-PK)) was then added to each sample as an internal standard. The quantification was conducted on a Thermo Fisher Scientific Dionex Ultimate 3000 UHPLC system coupled with a TSQ Quantiva Triple Quadrupole mass spectrometer (Thermo Fisher Scientific). Chromatographic separation was accomplished on an Agilent Zorbax StableBond C18 column (0.5 × 150 mm, 5 μm). The mobile phase consisted of 0.1% formic acid in water (A) and 0.1% formic acid in acetonitrile (B) and was delivered at a flow rate of 0.7 mL/min. A linear gradient elution program, beginning with 5% of mobile phase B, then increasing mobile phase B linearly from 5% to 70% over 25 min was employed. Both biotin and biotin-$d_4$ eluted at 6.8 min. The mass spectrometer was equipped with an ESI interface operated in positive ion mode. Quantification was accomplished in multiple reaction monitoring modes by monitoring a transition pair of $m/z$ 245.1 and 227.1 for biotin and 249.1 and 231.2 for the internal standard, biotin-$d_4$. The biotin level was calculated by fitting the area under the curve of

samples with a biotin standard curve, which was run in parallel for each analysis.

### UV spectroscopic studies

In the anaerobic chamber, 200 μM BioB.tb with or without 200 μM BsaP were prepared in 200 mM Tris, pH 8.5 to provide a final volume of 2 mL. Next, 5 mM DTT, 800 μM $Na_2S$, and 800 μM ferrous ammonium sulfate/ferric ammonium sulfate were consecutively added to the solution to reconstitute the protein. Prior to UV-vis analysis, the protein solution was buffer exchanged into 25 mM Tris, pH 8.0, 100 mM NaCl using Sephadex G-25 column (Cytiva, cat.17085101). The UV-vis spectra were obtained by scanning wavelengths from 350 nm to 800 nm (Agilent, model. Cary 100 UV-Vis).

### Bioinformatic analyses

BsaP homologs were identified using Blastn[45] searches. Full gene sequences for the BsaP hits were obtained from BV-BRC[46,47] and the dataset was filtered for genes located within 1 kb of the BioB open reading frame. A multiple sequence alignment for the homolog dataset was performed using MUSCLE[48]. A model was selected and a maximum likelihood phylogenetic tree was inferred using IQTree[49–51]. The tree was pruned using Treemmer[52], to reduce the dataset without loss in diversity. Subsequently, a new model was selected and tree reconstruction was performed, also using IQTree. Maximum likelihood tree construction was performed using a TVM + F + I + G4 model with 1000 bootstrap supports. The tree was rooted using minimal ancestor deviation[53]. The tree figure was drawn with ETE3[54], and the consensus sequence was identified using EMBOSS[55]. Alignments were visualized using pyMSAviz[56].

Full protein sequences for BioB hits were obtained by pulling BioB annotated sequences for all representative bacterial genomes from BV-BRC using BV-BRC CLI tools. To identify BsaP-dependent BioB homologs, Blastn search was run for all species and filtered for a BsaP homolog of at least 79 amino acids in length, with >60% identity, and with open reading frame within 1 kb of BioB open reading frame. Species without a BsaP homology hit were classified as BsaP-independent, and downsampled to 1000 species. Subsequently, multiple sequence alignments were performed using MUSCLE, and a phylogenetic tree was inferred using IQTree. The dataset was then pruned using Treemer down to 198 BsaP-independent BioB sequences, and 81 BsaP-dependent BioB sequences. MUSCLE was used to generate a multiple sequence alignment of the pruned dataset, and a phylogenetic tree was inferred with IQTree using an LG + F + I + G4 model with 1000 bootstrap supports. The tree was rooted with minimal ancestor deviation. BsaP-dependent and -independent BioB consensus sequences were identified using EMBOSS. Multiple sequence alignments were visualized using pyMSAviz. All source code is publicly available (https://github.com/nataliet87/bsaP_homolog_analyses).

Alphafold. Mycobacterial protein structure predictions were obtained from the Alphafold repository on May 20, 2022. The models used can be accessed at https://alphafold.ebi.ac.uk/entry/P9WPQ7, https://alphafold.ebi.ac.uk/entry/A0QX70, and https://alphafold.ebi.ac.uk/entry/P9WLT7.

### Reporting summary

Further information on research design is available in the Nature Portfolio Reporting Summary linked to this article.

## Data availability

The whole genome sequencing data generated in this study have been deposited in the BioSample database under accession codes SAMN30984225 [https://www.ncbi.nlm.nih.gov/sra?LinkName= biosample_sra&from_uid=30984225], SAMN30984226 [https://www. ncbi.nlm.nih.gov/sra?LinkName=biosample_sra&from_uid=30984226], SAMN30984221 [https://www.ncbi.nlm.nih.gov/sra?LinkName=

biosample_sra&from_uid=30984221]. The Tn libraries sequencing data generated in this study have been deposited in the NCBI Sequence Read Archive (SRA) under accession codes SRR22070226 to SRR22070233 for *M. tuberculosis* and SRR22069860 to SRR22069863 for *M. smegmatis*.
SRR22070226 [https://www.ncbi.nlm.nih.gov/sra/SRX18050633[accn]],
SRR22070227 [https://www.ncbi.nlm.nih.gov/sra/SRX18050632[accn]],
SRR22070228 [https://www.ncbi.nlm.nih.gov/sra/SRX18050631[accn]],
SRR22070229 [https://www.ncbi.nlm.nih.gov/sra/SRX18050630[accn]]
SRR22070230 [https://www.ncbi.nlm.nih.gov/sra/SRX18050629[accn]],
SRR22070231 [https://www.ncbi.nlm.nih.gov/sra/SRX18050628[accn]],
SRR22070232 [https://www.ncbi.nlm.nih.gov/sra/SRX18050627[accn]],
SRR22070233 [https://www.ncbi.nlm.nih.gov/sra/SRX18050626[accn]],
SRR22069860 [https://www.ncbi.nlm.nih.gov/sra/SRX18050277[accn]],
SRR22069861 [https://www.ncbi.nlm.nih.gov/sra/SRX18050276[accn]],
SRR22069862 [https://www.ncbi.nlm.nih.gov/sra/SRX18050275[accn]],
SRR22069863 [https://www.ncbi.nlm.nih.gov/sra/SRX18050274[accn]].
Sequencing data, codes and Alphafold models are available as described in the methods section. Source data are provided with this paper.

## Code availability

The code used for this work is available from https://github.com/nataliet87/bsaP_homolog_analyses, which is archived in Zenodo with the identifier https://doi.org/10.5281/zenodo.10912991[57].

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

## Acknowledgements
This work was supported by NIH grants AI143575 (SE, JSC, and DS) and AI143784 (CCA and DS).

## Author contributions
DQ, LB, SWP, HNL, SE, and DS designed microbiologic and genetic experiments, which were performed by DQ, LB, SWP, HNL, and JB. CCA and PG designed biochemical experiments, which were performed by PG. NT performed bioinformatic analyses. IVK and JCS performed structural analyses. DQ, PG, IVK, CCA, and DS wrote the manuscript with input from all authors.

## Competing interests
The authors declare no competing interests.
