## [Peer Review File · Nature Communications]

Mycobacterial biotin synthases require an auxiliary protein to convert dethiobiotin into biotinReviewer #1 (Remarks to the Author):

This is an excellent ms. The authors have thoroughly documented that Mtb BioB requires the BsaP protein for function in both Mtb and *M. smegmatis*. This is true even when Mtb BioB is used to complement an *E. coli* BioB deletion strain. It seems likely that BsaP contributes to building or restoration of the Mtb BioB auxiliary cluster.

Given the similarity between BioB and the LipA lipoyl synthase reactions, I was surprised that the authors did not discuss the successful restoration of the auxiliary cluster of LipA (and the mammalian LIAS homologue).

An item that the authors should consider is the Type II BioB found in obligately anaerobic bacteria in which the auxiliary cluster is an [Fe₄-S₄] cluster. The present authors seem likely to be unaware of Type II BioBs. This reviewer has known of these enzymes for >5 years but this was encumbered by proprietary issues. However, I stumbled on this thesis in Google Scholar

https://backend.orbit.dtu.dk/ws/portalfiles/portal/269335625/Afhandling_dl_h.pdf

Synthetic biology to address complex bottleneck challenges in high-value bioprocesses Ph.D.

Thesis by David Lennox-Hvenekilde

Technical University of Denmark (DTU)

2021

(first page of the thesis)

Users may download and print one copy of any publication from the public portal for the purpose of private study or research.

You may not further distribute the material or use it for any profit-making activity or commercial gain

You may freely distribute the URL identifying the publication in the public portal.

Hence the cat is officially out of the bag. This may not be germane to the present paper but UV spectra should detect a [Fe₂-S₂] cluster in the Mtb and *M. smegmatis*. In Fig 5b I don't see the pronounced 450 nm peak expected for a [Fe₂-S₂] cluster. Another good peak is at 330 nm but the figure may not include that region

Reviewer #2 (Remarks to the Author):

In this manuscript, the authors identified that the gene rv1590 in *Mycobacterium tuberculosis* is required to synthesize biotin. The authors first use genetic complementation studies to demonstrate that rv1590 is required for the conversion of dethiobiotin to biotin, the final step in the biotin biosynthetic pathway, which is catalyzed by BioB. The authors next expressed and purified the gene encoding this biotin synthase auxiliary protein (BsaP) and were able to show that BsaP is required for BioB activity in vitro. The identification of this protein is novel and the characterization of BsaP is useful for the development of new antibacterials that target the biotin biosynthetic enzymes. Although the genetic complementation studies appears to be extensive, I think the manuscript is lacking in biochemical and/or structural characterization of the role of BsaP in the BioB catalyzed reaction. Therefore, I don't think that the manuscript in its current form is ready for publication. I have listed suggestions and specific comments below.

Suggestions:

The authors have only used a single experiment to investigate the Fe-S cluster of MtBioB – BsaP. I think another method is also required given that one of the main findings is that the BsaP seems to be important for Fe-S cluster formation. The authors conclusions are that the BsaP protein is important for the formation of a 2Fe-2S cluster. However, I don't think the data in this manuscript is conclusive.

Have the authors considered using EPR to characterize the Fe-S cluster. EPR could be used to determine whether the Fe-S cluster is [2Fe-2S] or [3Fe-4S] etc.

Did the authors consider monitoring the change in absorbance as biotin is formed. A similar

experiment was described in ref 29, where the authors observed the change in absorbance at 460 nm as biotin was formed. This could be used to show the loss of the [2Fe-2S] cluster as the sulfur atom is transferred to make biotin.

Have the authors attempted to crystalize the Mtb BsaP protein or in complex with Mtb BioB?

The authors also attempted to make several mutants of Mtb BsaP, however they were unstable and therefore could not be investigated in the biochemical assays. Have the authors considered making a Mtb BioB-BsaP fusion protein, where the 2 domains are linked by a flexible linker, this could improve the stability and be useful for biochemical assays.

Specific comments:

Introduction:

Line 33 - Should be Mycobacterium tuberculosis

Line 35 - envelope?

Line 37 - biotin protein ligase

Line 65 - in vivo and in vitro

Line 67 - in vitro

Results:

Figure 4b-d -There are no error bars for any of the plots and there is also no standard error in the reported K_m , k_{cat} or K_d values. These should be added and the relevant details should be added to the figure legend. In addition, the units for K_m and K_d are missing.

Is Acidomycin a competitive inhibitor of MtB BioB? The authors have not demonstrated this in this manuscript. This information is needed in order to calculate a K_i .

There are no error bars in extended data figure 6 and the standard error is not reported for the IC_{50} and K_i values.

Figure 5b - The wavelength indicated by the dashed lines should be listed in the figure legend.

Figure 5c - spelling error in y-axis: should be initial not incial

Figure 6b - EcBioB Cys188 and Arg260 (and the corresponding MtBioB residues are not labeled)

Extended data figure 7 - I think there are some details missing in the figure legend. What do the open and closed circles represent (+/- biotin?). What is HA?

Methods:

The full Western blots are not shown in the supplementary/extended data

The details for the immunoblots are not listed - what were the primary antibodies, the dilution factors for all antibodies, timing, temperature etc.

There are also no details for the inhibition assays and the determination of the K_d . This includes enzyme concentration, inhibitor concentration and how the data was fitted.

Reviewer #1 (Remarks to the Author):

This is an excellent ms. The authors have thoroughly documented that Mtb BioB requires the BsaP protein for function in both Mtb and *M. smegmatis*. This is true even when Mtb BioB is used to complement an *E. coli* BioB deletion strain. It seems likely that BsaP contributes to building or restoration of the Mtb BioB auxiliary cluster.

We are grateful to the reviewer for this evaluation.

Given the similarity between BioB and the LipA lipoyl synthase reactions, I was surprised that the authors did not discuss the successful restoration of the auxiliary cluster of LipA (and the mammalian LIAS homologue).

We thank the reviewer for pointing this out and agree that it will be interesting to analyze the importance of BsaP for other enzymes, such as LipA. Addressing this experimentally is beyond the scope of this manuscript, but we added this point to the discussion.

An item that the authors should consider is the Type II BioB found in obligately anaerobic bacteria in which the auxiliary cluster is an [Fe4-S4] cluster. The present authors seem likely to be unaware of Type II BioBs. This reviewer has known of these enzymes for >5 years but this was encumbered by proprietary issues. However, I stumbled on this thesis in Google Scholar

https://backend.orbit.dtu.dk/ws/portalfiles/portal/269335625/Afhandling_dl_h.pdf
Synthetic biology to address complex bottleneck challenges in high-value bioprocesses Ph.D. Thesis by David Lennox-Hvenekilde
Technical University of Denmark (DTU)
2021

Users may download and print one copy of any publication from the public portal for the purpose of private study or research.

You may not further distribute the material or use it for any profit-making activity or commercial gain

You may freely distribute the URL identifying the publication in the public portal.

Hence the cat is officially out of the bag.

Thank you very much for pointing this out. Referring to the elegant studies recently published by Lachowicz et al., we included type II biotin synthases to both the introduction and the discussion of the revised manuscript.

This may not be germane to the present paper but UV spectra should detect a [Fe2-S2] cluster in the Mtb and *M. smegmatis*. In Fig 5b I don't see the pronounced 450 nm peak expected for a [Fe2-S2] cluster. Another good peak is at 330 nm but the figure may not include that region.

We thank the reviewer for their careful analysis of the UV spectra. Due to interference by inorganic iron and sulfur salts, we were unable to record the spectrum down to 330 nm. The blue trace of aerobically purified BioB does show a pronounced peak at 452 nm expected for a [2Fe-2S]²⁺ cluster. Upon chemical reduction in the presence of iron and sulfide the UV spectra (illustrated in the orange and red traces) has a dramatic change and the spectra reflects overlapping spectra from both [2Fe-2S]²⁺ and [4Fe-4S]²⁺ clusters. As reported for the *E. coli* BioB (Biochemistry 2001, 40, 8352-8358) we also observe broad shoulders 460 nm from the [2Fe-2S]²⁺ and a peak at 410 nm consistent with the [4Fe-4S]²⁺. The A₄₂₀/A₄₅₂ ratio of the fully reconstituted BioB provides information on the ratios of the two Fe-S clusters and Jerrett has shown with *Ec*BioB that a value of 1.18 corresponds to 1:1 ratio of [2Fe-2S]²⁺ and [4Fe-4S]²⁺ (Jarrett, JT et al. Purification, Characterization, and Biochemical Assays of Biotin Synthase From *Escherichia coli* *Methods Enzymol.* **2018**, 606, 363-388). Anaerobic reconstitution of *Mt*BioB+Bsap gave a A₄₂₀/A₄₅₂ (1.53/1.31) = 1.17 further supporting formation of a [2Fe-2S]²⁺ cluster.

Reviewer #2 (Remarks to the Author):

In this manuscript, the authors identified that the gene *rv1590* in *Mycobacterium tuberculosis* is required to synthesize biotin. The authors first use genetic complementation studies to demonstrate that *rv1590* is required for the conversion of dethiobiotin to biotin, the final step in the biotin biosynthetic pathway, which is catalyzed by BioB. The authors next expressed and purified the gene encoding this biotin synthase auxiliary protein (BsaP) and were able to show that BsaP is required for BioB activity *in vitro*. The identification of this protein is novel and the characterization of BsaP is useful for the development of new antibacterials that target the biotin biosynthetic enzymes. Although the genetic complementation studies appears to be extensive, I think the manuscript is lacking in biochemical and/or structural characterization of the role of BsaP in the BioB catalyzed reaction. Therefore, I don't think that the manuscript in it's current form is ready for publication. I have listed suggestions and specific comments below.

We thank the reviewer for this careful evaluation of our manuscript. We agree that the discovery of BsaP should lead to additional biochemical and structural experiments, including those mentioned by reviewer 1. However, we believe that our findings are conclusive even in the absence of such additional experiments. We carefully reviewed our main conclusions and, where appropriate, edited them to assure that they are supported by what we believe to be sufficient experimental evidence.

Suggestions:

The authors have only used a single experiment to investigate the Fe-S cluster of MtBioB – BsaP. I think another method is also required given that one of the main findings is that the BsaP seems to be important for Fe-S cluster formation. The authors conclusions are that the BsaP protein is important for the formation of a 2Fe-2S cluster. However, I don't think the data in this manuscript is conclusive.

As mentioned above, we respectfully disagree with this evaluation and believe that our main conclusions – that enzymatic activity of mycobacterial biotin synthases requires BsaP – is both novel and supported by sufficient experimental evidence.

The main finding of this work - that the enzymatic activity of mycobacterial biotin synthases requires BsaP – is conclusively supported by sufficient experimental evidence in cells and biochemically. We do not claim to have shown that BsaP is important for BioB 2Fe-2S cluster *formation*. Based on the sequence comparison between BsaP independent and BsaP dependent BioB proteins, we hypothesize that BsaP is likely involved in the part of the reaction supported by 2Fe-2S cluster. At this point, we do not know the mechanism by which BsaP assists in BioB reaction, and we do not make any claims that we do. We believe that these are all exciting new avenues for future studies.

Have the authors considered using EPR to characterize the Fe-S cluster. EPR could be used to determine whether the Fe-S cluster is [2Fe-2S] or [3Fe-4S] etc.

We agree that further mechanistic studies using paramagnetic resonance and Mössbauer spectroscopy would be needed to fully elucidate the mechanism. However, these experiments are complicated by the need to use a strong reductant to reduce the clusters to the paramagnetic 1+ forms and the presence of [2Fe-2S]²⁺ and [4Fe-4S]²⁺ clusters in BioB, which leads to two species (Biochemistry, 2011, 50, 7953-7963; Biochemistry 2004, 43, 2022) that have similar g-values and relaxation as [2Fe-2S]²⁺ clusters, but the distinction between these species is unknown. Another manuscript focused exclusively on the characterization of the BioB [2Fe-2S]²⁺ cluster with limited success (Biochem. Biophys Res. Commun. 2009, 381, 487-490). We believe the EPR characterization should be included in a subsequent manuscript focused exclusively on the functional and structural characterization of BioB and BsaP.

Did the authors consider monitoring the change in absorbance as biotin is formed. A similar experiment was described in ref 29, where the authors observed the change in absorbance at 460 nm as biotin was formed. This could be used to show the loss of the [2Fe-2S] cluster as the sulfur atom is transferred to make biotin.

We used a LC-MS/MS assay with an authentic deuterated biotin-d4 standard to rigorously measure the reaction rate. Since our UV-Vis trace of Mtb BioB is nearly identical to the reported E. coli BioB, we did not seek to further confirm the [2Fe-2S]²⁺ cluster by loss of 460 nm absorbance in UV-Vis. We would expect to observe a decrease at 460 nm as reported for E. coli BioB in reference 33.

Have the authors attempted to crystalize the Mtb BsaP protein or in complex with Mtb BioB?

We have not attempted to crystalize Mtb BsaP, alone or in complex with Mtb BioB. We agree with the reviewer that it will be fascinating to determine these structures and we are seeking a collaborator to pursue these studies. But we believe that these experiments are beyond the scope of our manuscript and not required to support our main conclusions.

The authors also attempted to make several mutants of Mtb BsaP, however they were unstable and therefore could not be investigated in the biochemical assays. Have the authors considered making a Mtb BioB-BsaP fusion protein, where the 2 domains are linked by a flexible linker, this could improve the stability and be useful for biochemical assays.

We have not attempted to generate fusion proteins. We agree that the reviewers that this will be an approach worthwhile pursuing as BsaP is characterized further. However, we don't believe that experiments with fusion proteins are required to support our main conclusions.

Specific comments:

Introduction:

Line 33 - Should be Mycobacterium tuberculosis

Good point. Thank you. This has been edited as requested.

Line 35 – envelope?

Thank you. This has been corrected as suggested.

Line 37 – biotin protein ligase

Thank you. This has been corrected as suggested.

Line 65 – in vivo and in vitro

Line 67 – in vitro

The section corresponding to lines 65/67 of the original manuscript (lines 69-73 of the revised manuscript) has been edited for clarity.

Results:

Figure 4b-d -There are no error bars for any of the plots and there is also no standard error in the reported K_m , k_{cat} or K_d values. These should be added and the relevant details should be added to the figure legend. In addition, the units for K_m and K_d are missing.

Error bars have been added to Figures 4a-d as requested, key assay details have been included in the figure legend, and units for the reported K_m , k_{cat} , and K_D values have been added.

Is Acidomycin a competitive inhibitor of MtB BioB? The authors have not demonstrated this in this manuscript. This information is needed in order to calculate a K_i .

The reviewer is correct, we cannot refer to the inhibition constant as a " K_i ," without performing inhibition studies at different substrate concentrations. We have thus corrected the figure to clearly

indicate this is an “apparent K_i ” or “ $K_{i,app}$ ” and this number is reported in the main manuscript. We previously demonstrated acidomycin exhibits competitive inhibition with respect to DTB using *E. coli* BioB (reference 29, PMID: 30652474), thus we can calculate a true K_i using the Cheng-Prusoff equation as shown in the figure inset, which is close to the experimentally determined $K_{i,app}$ since the inhibition assays were performed at 10 μ M near our calculated K_M of DTB (7 μ M).

The extended Figure 6 panel ‘a’ shows the fit of the concentration-response plot to the Hill equation to determine the IC_{50} . The data is separately fit to the Morrison equation in panel ‘b’ since we observed tight-binding inhibition (when the $IC_{50} \sim [E]$). We use the term ‘ $K_{i,app}$ ’ following the nomenclature recommended by Robert Copeland (*Evaluation of Enzyme Inhibitors in Drug Discovery: A Guide for Medicinal Chemists and Pharmacologists*, 2nd Edition, ISBN: 978-1-118-48813-3).

There are no error bars in extended data figure 6 and the standard error is not reported for the IC_{50} and K_i values.

Error bars were added to the reported IC_{50} and K_i values in extended Figures 6.

Figure 5b - The wavelength indicated by the dashed lines should be listed in the figure legend.

Thank you for pointing this out. This information has been added.

Figure 5c – spelling error in y-axis: should be initial not inial

Thank you. The spelling error has been corrected..

Figure 6b - EcBioB Cys188 and Arg260 (and the corresponding MtBioB residues are not labeled)

In the original version of this figure we only labeled the residues that are different in BioB.ec and BioB.tb. In the revised figure the residues requested by the reviewer have also been labeled.

Extended data figure 7 – I think there are some details missing in the figure legend. What do the open and closed circles represent (+/- biotin?). What is HA?

The open and closed symbols represent +/- biotin. This is indicated in the legend below panel A. HA stands for hemagglutinin tag. This information has been added to the figure legend.

Methods:

The full Western blots are not shown in the supplementary/extended data

Full Western blots have been added as Extended Data Fig. 11.

The details for the immunoblots are not listed – what were the primary antibodies, the dilution factors for all antibodies, timing, temperature etc.

The requested information has been added to the methods.

There are also no details for the inhibition assays and the determination of the K_d . This includes enzyme concentration, inhibitor concentration and how the data was fitted.

We have now provided further details on the inhibition assays including enzyme concentration, inhibitor concentrations used along with the equations used to fit the data for determination of $K_{i,app}$ and K_D for the BioB-BsaP interaction.